# Insights into the Replication Kinetics Profiles of Malaysian SARS-CoV-2 Variant Alpha, Beta, Delta, and Omicron in Vero E6 Cell Line

**DOI:** 10.3390/ijms251910541

**Published:** 2024-09-30

**Authors:** Zarina Mohd Zawawi, Jeevanathan Kalyanasundram, Rozainanee Mohd Zain, Adiratna Mat Ripen, Dayang Fredalina Basri, Wei Boon Yap

**Affiliations:** 1Virology Unit, Infectious Disease Research Centre, Institute for Medical Research, National Institutes of Health, Ministry of Health, Shah Alam 40170, Malaysia; jeevan@moh.gov.my (J.K.); rozainanee@moh.gov.my (R.M.Z.); 2Center for Toxicology and Health Risk Studies, Faculty of Health Sciences, Universiti Kebangsaan Malaysia, Kuala Lumpur 50300, Malaysia; 3Cancer Research Centre, Institute for Medical Research, National Institutes of Health, Ministry of Health, Shah Alam 40170, Malaysia; adiratna@moh.gov.my; 4Center for Diagnostic, Therapeutic and Investigative Studies, Faculty of Health Sciences, Universiti Kebangsaan Malaysia, Kuala Lumpur 50300, Malaysia; dayang@ukm.edu.my; 5One Health UKM, Universiti Kebangsaan Malaysia, Bangi 43600, Malaysia

**Keywords:** SARS-CoV-2, variants, replication kinetics, infectivity, mutations, cell culture adaptation

## Abstract

Comprehending the replication kinetics of SARS-CoV-2 variants helps explain why certain variants spread more easily, are more contagious, and pose a significant health menace to global populations. The replication kinetics of the Malaysian isolates of Alpha, Beta, Delta, and Omicron variants were studied in the Vero E6 cell line. Their replication kinetics were determined using the plaque assay, quantitative real-time PCR (qRT-PCR), and the viral growth curve. The Beta variant exhibited the highest replication rate at 24 h post-infection (h.p.i), as evidenced by the highest viral titers and lowest viral RNA multiplication threshold. The plaque phenotypes also varied among the variants, in which the Beta and Omicron variants formed the largest and smallest plaques, respectively. All studied variants showed strong cytopathic effects after 48 h.p.i. The whole-genome sequencing highlighted cell-culture adaptation, where the Beta, Delta, and Omicron variants acquired mutations at the multibasic cleavage site after three cycles of passaging. The findings suggest a strong link between the replication rates and their respective transmissibility and pathogenicity. This is essential in predicting the impacts of the upcoming variants on the local and global populations and is useful in designing preventive measures to curb virus outbreaks.

## 1. Introduction

Severe acute respiratory syndrome coronavirus 2 (SARS-CoV-2) is the causative agent responsible for the COVID-19 pandemic causing millions of deaths and morbidity globally, including Malaysia. It belongs to the *Betacoronavirus* family and is closely related to the SARS-CoV virus that was first documented in November 2002–2003 [1]. Malaysia reported the emergence of Alpha and Beta variants during the third wave of the pandemic, whereas the Delta (B.1.617.2) variant dominated the fourth wave but later was replaced by AY.59 and AY.79 in Peninsular Malaysia and AY.23 in East Malaysia. The Omicron variant (BA.1 lineage) emerged in December 2021 during the fifth wave [2,3]. The mutational rate of SARS-CoV-2 was estimated to be around 1 × 10^−6^ to 2 × 10^−6^ mutations per site per generation [4]. This rapid mutation is attributed to factors such as lack of proofreading by the viral RNA-dependent RNA polymerase (RdRp) and selective pressures from the host immune response. The situation was worsened by the extensive global spread, creating abundant opportunities for the accumulation of virus particles in infected individuals, which in turn resulted in the successful evolution of SARS-CoV-2 [5]. This genetic diversity subsequently led to the emergence of variants with distinctive phenotypes characteristics, and increased transmissibility, disease severity, and immune evasion.

Throughout the pandemic, the wild-type SARS-CoV-2 virus has evolved to form numerous pathogenic and contagious variants. Despite the complexity of the viral evolution, certain general insights have been deduced. The Alpha variant (B.1.1.7), identified in the United Kingdom in September 2020, is primarily characterized by the N501Y mutation along with the common D614G mutation in the S protein, hence the increased transmissibility [6]. In December 2020, the Beta variant (B.1.351) emerged in South Africa with a number of significant mutations (N501Y, E484K, and K417N) that affected the binding of neutralizing antibodies, therefore conferring resistance to monoclonal antibodies [7,8]. The Delta variant (B.1.617.2), on the other hand, emerged in India in late 2020 or early 2021 and was shown to be 40–60% more transmissible. The Delta variant carried a few mutations (L452R, P681R, and T478K) that impacted the host immunity, viral entry, infectivity, and transmissibility [9]. The Omicron variant (B.1.1.529) emerged in South Africa in November 2021 and possessed over 60 mutations. More than half of the mutations were found in the S protein, including key mutations in the receptor-binding domain (RBD) such as ∆69–70 deletion, T95I, G142D/∆143–145 deletion, K417N, T478K, N501Y, N655Y, N679K, and P681H [10]. The mutations modified the structure of the S protein and its RBD, and deteriorated the neutralizing antibody binding. As a result, the virus transmissibility and infectivity, and its ability to evade host immunity were heightened compared to the Delta variant.

The impact of viral genome mutations on viral transmissibility and infectivity can be reflected in the virus replication profile [11]. An efficient virus replication promotes viral loads in bodily fluids, which in turn facilitates virus shedding and transmission. Moreover, the higher viral load also can induce extensive tissue damage and a stronger immune response, and leads to increased pathogenicity [12]. Therefore, dissecting the replication kinetics of various Malaysian SARS-CoV-2 variants is crucial for assessing the impacts of the virus infections on patients, as well as serving as a guide for improving the control and preventive measures of COVID-19. To better understand the relationship between the replication kinetics of SARS-CoV-2 Alpha, Beta, Delta, and Omicron variants and their pathogenicity, the virus infections were performed in the African Green Monkey-derived Vero E6 kidney cell line in this study. The Vero E6 cell line has been widely used in coronavirus research as it is highly susceptible to coronavirus infections such as SARS-CoV and SARS-CoV-2, and therefore supports coronavirus replications and enables high viral load synthesis [13,14]. Although lacking the TMPRSS2 protease required for virus entry, Vero E6 cells express the angiotensin-converting enzyme 2 (ACE2) receptor for SARS-CoV-2 attachment [15]. In addition, their deficiency in interferon production allows robust viral replication without host antiviral interference [16]. Altogether, the Vero E6 cell line is an eligible host cell model for studying and comparing the replication dynamics of SARS-CoV-2 variants.

Rosli et al. (2023) investigated the growth profiles of Malaysian SARS-CoV-2 strains (clades L/Lineage B Wuhan, GR/Lineage B.1.1.354, and O/Lineage B.6.2) collected in the early stages of the COVID-19 pandemic in various mammalian cell lines, including Vero E6, Vero CCL-81, and Calu-3 [17]. Although the growth profiles of the variants were reportedly cell line-dependent, the study lacked a clear and direct comparison between the studied variants. Given that the replication kinetics and virus infectivity are highly associated with the unique mutations in the spike proteins of the individual variants, this study aimed to investigate and compare the replication profiles of Malaysian strains of SARS-CoV-2 Alpha, Beta, Delta, and Omicron variants in the Vero E6 cell line.

## 2. Results

### 2.1. Morphology Changes of Vero E6 Cells Infected with Malaysian SARS-CoV-2 Variants

All four SARS-CoV-2 variants (Alpha, Beta, Delta, and Omicron) demonstrated cytopathic effects (CPEs) through cell lysis in Vero E6 cells. The CPEs started to appear at 24 h.p.i. and became more severe after 48 h.p.i. [Figure 1a]. At 48 h.p.i., approximately 70–80% of the cells were rounded, detached, and lysed. By 72 h.p.i., all infected cells were lysed and floating on the surface of the media.

### 2.2. Viral RNA Multiplication of SARS-CoV-2 Variants in Vero E6 Cells

Vero E6 cells were infected with the SARS-CoV-2 variants at MOI 0.001. The Ct values of the variants harvested at designated post-infection time points were determined using qRT-PCR up to 72 h.p.i. Figure 1b shows that the viral loads of the variants increased in a time-dependent manner. Among the four SARS-CoV-2 variants, the Beta variant exhibited significantly higher replication efficiency than the other variants as indicated by the lowest Ct values, especially that of 24 h.p.i. Meanwhile, the Omicron variant recorded the highest Ct value at 24 h.p.i. The Ct values of all variants started to plateau from 48 to 72 h.p.i.

### 2.3. Plaque Formation and Phenotypes

Figure 2 illustrates the plaque phenotypes and replication kinetics of the four SARS-CoV-2 variants. In the plaque assay [Figure 2a], it was shown that the Beta variant formed the largest plaques, followed by the Alpha and Delta variants. The Omicron variant produced the smallest plaques as seen in the well. Visible plaques were observed on day 3 (72 h) p.i with the Alpha, Beta, and Delta variants; nonetheless, the plaque formation was delayed for 24 h with the Omicron variant. Unlike the other variants whose plaque phenotypes were relatively consistent, the Alpha variant demonstrated a rather mixed plaque phenotype with both large and small plaques.

### 2.4. Replication Kinetics of SARS-CoV-2 Variants in Vero E6 Cells

Subsequently, the viral titers of all SARS-CoV-2 variants were determined in Vero E6 cells. By comparing the viral titers at the designated time points [Figure 2b], the replication patterns of the variants were almost similar. At 24 h.p.i, the replication of all variants increased drastically. The Beta variant already exhibited a significantly higher virus titer (*p* < 0.002) than the other variants at 24 h.p.i. The viral titers peaked at 48 h.p.i., indicating the maximum viral replication rates. Following the peak, declining viral titers were observed, continuing through to 96 h.p.i. This trend suggests a dynamic interaction between the viral replication and host cellular responses, leading to the reduction of viral loads. Collectively, the results imply that the distinctive replication patterns are variant-specific, and the virus replication dynamics could change over time.

### 2.5. Whole-Genome Sequencing of SARS-CoV-2 Alpha, Beta, Delta, and Omicron Variants

Given that mutations in the S protein of SARS-CoV-2 can severely impact the virus transmissibility and pathogenicity, next-generation sequencing (NGS) was conducted using the Oxford Nanopore GridION system to determine the genome stability of the variants after three cycles of passaging. The whole-genome sequences of the variants were aligned and compared with their original sequences [Table 1] using Nextclade v3.8.2.

The results revealed that the Beta, Delta, and Omicron variants exhibited cell culture adaptations in which deletion mutations were detected at the S1/S2 boundary [Figure 3a,b]. Several point mutations either by deletion or substitution were found in the multibasic cleavage sites (MBCSs) of the S proteins: (i) Beta—N679del, S680del, P681del, R683del, A684del, and R685del; (ii) Delta—Q677del, T678del, N679del, and S680del; and (iii) Omicron—R682W [Figure 3b]. The Alpha variant appeared to be relatively stable as there were no additional mutations identified in the variant’s MBCSs after three cycles of passaging.

## 3. Discussion

Similar to the other global variants, Malaysian isolates of SARS-CoV-2 variants possibly exhibit differences in their replication kinetics [18]. It is generally accepted that variants with higher abilities in synthesizing infective virions are associated with greater transmissibility and infectivity [19]. In this study, four variants, namely Alpha, Beta, Delta, and Omicron, sampled in Malaysia during the COVID-19 pandemic were studied and compared. The Gamma variant was not included in this study as it was not a circulating variant in Malaysia [2]. Notably, in Vero E6 cells, all SARS-CoV-2 variants exhibited strong cytopathic effects (CPEs) after 48 h.p.i. in the form of cell lysis. The severity of CPEs became more visible over the incubation period, particularly at 48 h.p.i. due to extensive virus-induced cell damage; this limited the replication of the variants as implied by the constant Ct values and declining viral titers at 48–72 and 48–96 h.p.i., respectively. In this study, the viral titers were monitored up to 96 h.p.i. to better decipher the replication patterns of the variants, which increased at 24 h.p.i., culminated at around 48 h.p.i., and subsequently declined at 72–96 h.p.i., thereby providing insights into the duration of infectivity. Monitoring the viral RNA multiplication thresholds beyond 72 h.p.i. did not provide additional information on the replication dynamics of the variants as their RNA levels had already plateaued by that point.

The plaque size can serve as an indication of the transmissibility or contagiousness of the variants. The variants exhibited plausible variations in the plaque size, which was correlated with the RNA multiplication threshold and virus titer of the variants. In this study, the Beta variant formed the largest plaques, and yielded the highest number of RNA multiplication (lowest Ct value) and virus titers at 24 and 48 h.p.i. This suggests that the Beta variant demonstrated a higher viral replication rate and infectivity [20]. Similar findings were also reported by several previous studies [21,22,23] comparing the replication kinetics of the Beta variant to that of the Alpha variant and a few other variants. The Beta variant was also reported to form larger plaques and demonstrate higher thermal stability, hence an increased viral replication rate. The greater replication and infectivity of the Beta variant was linked to its E484K mutation in the S protein, which caused reduced antibody neutralization, and A701V mutation that impaired the attachment of the S protein to the ACE2 receptor [24]. As a result, those unique mutations (E484K and A701V) were likely to provide a fitness advantage to the Beta variant, making it globally dominant before the emergence of the later variants. This is in line with Theobald Smith’s Law of Declining Virulence. The theory proposed that infectious entities tend to evolve into less virulent variants to improve their transmission [25]. The compromised virulence prevents the occurrence of over-reactive immune responses in host cells, thereby ensuring the gradual synthesis of new virus particles. Once the virions reach the transmissibility threshold [26], the variant reclaims its virulence and infectivity, and a new outbreak wave is observed. Whether the newly emerged variant causes more severe and deadly infections, it depends on factors such as pre-existing host immunity induced by vaccination or prior infections, and effective public health measures.

In contrast, the Alpha variant was shown to form smaller plaques with mixed phenotypes and exhibited relatively lower viral replication and infectivity than the other variants. The differences were remarkably evident compared to that of the Beta variant. This is in line with that reported by Jeong et al. [23], in which the Alpha variant also produced smaller plaques and fewer infectious viral particles and resulted in a lower intracellular viral RNA concentration than the Beta, Gamma, and Delta variants. However, in comparison with the wild-type (wt) SARS-CoV-2, the Alpha variant demonstrated an improved replication kinetic due to the presence of D614G and N501Y mutations [27] and impaired type-1 interferon activities by the P681H mutation in the S protein [28].

Similarly, the Delta variant also exhibited smaller plaque formation. This is consistent with the findings reported by Tanneti et al. [29], which showed that both the Alpha and Delta variants produced significantly smaller plaques in Vero E6 TMPRSS2 cells. In spite of the smaller plaque formation, the Delta variant still yielded significantly higher infectious viral titers at 24 and 48 h.p.i in this study. The relatively greater viral titers are somewhat similar to those observed in the real infections of the Delta variant in infected hosts [30,31]. It is suggested that the Delta variant might release newly synthesized viral particles without causing extensive cell damage, hence the increased virulence but less remarkable plaque size.

Despite being the currently circulating variant globally, the Omicron variant demonstrated slower replicative ability than the other studied variants in this study. The formation of visible plaques was also delayed (about 4 to 5 days) compared to its fraternity variants. Shuai et al. [32] discovered that the Omicron variant replicated less efficiently than the wt SARS-CoV-2, Alpha, Beta, and Delta and caused comparatively lower cellular damage. The slower replication kinetics of the Omicron variant can be attributed to several factors, among others, the altered entry pathway that favors endocytosis over the plasma membrane route [33]. In addition, the Omicron variant is also less effective in inhibiting the host cell’s innate immune response than the Delta variant. This explains its slower replication due to its relatively higher sensitivity to the antiviral immunity of host cells [34]. Collectively, the Omicron variant exhibits low fusogenic potential, which reduces cell-to-cell spread and impacts the efficiency of viral replication and virus spread [35]. This also explains why it took longer for the Omicron variant to form visible plaques than the Alpha, Beta, and Delta variants.

Reduced fusogenicity is associated with milder disease progression, as seen in Omicron-infected patients [36]. This phenomenon is linked to its unique H655Y [37] and N856K [38] mutations in the S protein. The H655Y mutation stabilizes the spike trimer conformation that is important for viral entry into host cells [37]. This in turn decreases the flexibility of the S protein through hydrophobic interactions with specific residues. The loss of the plasticity of the S protein interferes with the furin cleavage site and affects envelope-membrane fusion. According to Sun et al. [38], when the N856K substitution was introduced in the spike of Omicron BA.1, it was 9.2- and 11.9-fold less fusogenic than the Delta or D614G variant, respectively. The aforementioned H655Y and N856K mutations were also observed in the Omicron variant of this study [Table 1]. The substitutions are believed to potentially prevent the release of the spike protein and obstruct the conformational change of the spike, and this, therefore, explains the delayed plaque formation and lower viral titer due to the lower fusogenicity of the Omicron variant observed at 24 h.p.i.

Park et al. [39] investigated and compared the fusogenicity of the Omicron BA.1 variant with the other earlier lineages (Alpha, Beta, and Delta) and noted that the fusogenic activities were proportional to the infectivity of the variants. The study attributed the reduced fusogenicity and infectivity of Omicron BA.1 to the unique mutations in the S1 and S2 domains [39]. Similarly, in this study, Figure 3b shows that the Omicron variant exhibited substitution mutations of N679K and P681H in the S1/S2 domain, which were not present in the Alpha, Beta, and Delta variants. In addition, Park et al. [39] also identified and reported five additional mutations (T547K, H655Y, N856K, Q954H, N969K) in the viral S protein that were responsible for the reduced fusogenicity of Omicron BA.1. This finding is consistent with that listed in Table 1. It is, therefore, noteworthy that the presence of multiple mutations in the S protein of the Omicron variant, especially in the S1/S2 region, may reduce membrane fusion between the virions and host cells, and subsequently lower the impact of virus infection.

In this study, whole-genome sequencing was performed to investigate the genome stability and cell culture adaptability of the variants after three cycles of passaging. Their sequences were aligned and compared with that of the original clinical samples. The results revealed several amino acid deletion mutations at the multibasic cleavage sites (MBCSs) in the S proteins of the Beta and Delta variants, specifically at the furin cleavage site (^677^QTNSPRRAR^685^). The Omicron variant acquired only an amino acid substitution of R682W at the MBCS. Mutations at the MBCS can affect the entry mode of the variants, i.e., the endosomal pathway (slower) or the cell-surface pathway (faster) as the entry mode of choice is largely dependent on the proteolytic activities of the TMPRSS2 protein at the MBCS [40]. As a result, any mutations altering the proteolytic reactions of TMRPSS2 at the MBCS can eventually modify the fusogenicity of infectious virions with the host cell membrane. In addition, the frequency of TMPRSS2 present on the host cell surface also affects the mode of viral entry. When the number of TMPRSS2 is low or negligible, the virus executes the slower endosomal entry pathway and vice versa. Therefore, to accommodate the absence of TMPRSS2 protease on Vero E6 cells, SARS-CoV-2 variants swiftly adapt themselves through genome mutations, particularly at the furin cleavage site so that they can perform alternative endosomal entry. This adaptation enhances the viral fitness or replication in the absence of the TMPRSS2-mediated pathway [41].

Introducing novel amino acid motifs, such as the PRRA and QTQTN of the SARS-CoV-2 S protein into that of SARS-CoV drastically affects virus entry into host cells and impacts the infectivity and transmissibility of SARS-CoV [42]. This implies the importance of PRRA and QTQTN motifs in concerting the entry of SARS-CoV-2 variants as demonstrated in this study. Romeu (2023) also suggested that the acquired PRRA furin polybasic motif was a key evolutionary step leading to the emergence of the pandemic coronavirus. The PRRA of SARS-CoV-2 showed almost 100% similarity with that of the human RNA [43]. This scenario indicates a potential recombination event between the virus genome and human RNA.

In this study, the Beta and Delta variants exhibited amino acid deletions specifically at ^679^NSP-RAR^685^ and ^677^QTNS^680^, respectively, after three consecutive passages. Liu et al. [44] reported that residue deletions usually occurred after two cycles of passaging. The study found the QTQTN deletion in clinical samples and laboratory-adapted virus isolates, and the NSPRRAR deletion in the laboratory-adapted isolates using the Oxford Nanopore Technologies (ONT) platform. However, their empirical findings indicated that the NSPRRAR deletion was less likely to impact the virus replication in Vero and Vero-E6 cells; the QTQTN deletion, on the other hand, may restrict the late-phase viral replication [44]. The findings were somehow contradictory to those of Davidson et al. [45], in which mutations that altered or deleted the putative furin cleavage site could lead to a significant change in the virus replication. Considering those previous findings, it is speculated that deletions at those specific sites may have varying effects on the viral replication, both in cell culture and clinical samples. In this study, the effects of ^679^NSP-RAR^685^ deletion in the Beta variant were similar to that reported previously [21,22,23]. The mutation did not significantly affect the viral replication and infectivity as indicated by the plaque size and replication of the Beta variant. However, the smaller plaque formed by the Delta variant may be attributed to the ^677^QTNS^680^ deletion and it is consistent with that reported by Tanneti et al. [29]. The deletion likely disturbed the viral replication in the later stages as evidenced by the smaller plaque formation, which implied reduced viral growth and spread within the cell culture. To substantiate how the manipulation of amino acids in the S protein by SARS-CoV-2 variants affects virus adsorption, entry, and replication as a whole, further bioinformatic and empirical analyses are required.

Notably, cell culture adaptation may contribute to an overestimation of viral transmissibility based on the increased or decreased viral load and plaque size. Plus, due to the lack of interferon response in Vero E6 cells, larger plaques, higher viral titers and increased replication rates are likely to occur than in natural infections. In view of this, it is important to note that the enhanced viral replication in mammalian cell cultures may not accurately represent their inherent behaviors in natural hosts or reservoirs. In a nutshell, while these findings provide valuable insights into the replication kinetics of SARS-CoV-2 variants, interpreting viral transmissibility and pathogenicity in vitro requires rigorous and meticulous empirical investigations.

## 4. Materials and Methods

### 4.1. Cell Culture

The African green monkey (Vero E6; ATCC CRL-1586) cell line was used in this study. The cells were maintained in Dulbecco’s minimal essential medium (DMEM; Nacalai Tesque, Kyoto, Japan) supplemented with 1% penicillin–streptomycin (10,000 U/mL) (Nacalai Tesque, Kyoto, Japan), 25 mM HEPES, L-glutamine, and 10% heat-inactivated fetal bovine serum (FBS; Nacalai Tesque, Kyoto, Japan). The cells were passaged every 3 to 4 days by dissociating them with TrypLETM Express Enzyme (Gibco, Grand Island, NE, USA). The cells were incubated at 37 °C in 5% CO_2_.

### 4.2. Virus Propagation, Isolation, and Preparation of Virus Stocks

SARS-CoV-2 variants including Alpha (B.1.1.7, EPI-ISL-877228), Beta (B.1.351, EPI-ISL-4730383), Delta (AY.59, EPI-ISL-3425462), and Omicron (BA.1.1, EPI-ISL-11105858) were studied. The archived viruses were obtained from the Virology Unit, Institute for Medical Research (IMR), Malaysia Ministry of Health. They were isolated from COVID-19 patients’ nasopharyngeal swab samples. All four SARS-CoV-2 variants were confirmed by real-time reverse transcription polymerase chain reaction (qRT-PCR) and Next-Generation sequencing. Virus propagation and isolation were carried out in Vero E6 cells for three passages and stored as virus stocks. The plaque assay was performed to determine the multiplicity of infection (MOI) of the virus stocks and an MOI of 0.001 was used in all assays. The above experiments were done in the Biosafety Level 3 facility, IMR.

### 4.3. Plaque Assay

The titration of viral load in the infectious culture supernatants collected at five different time points was conducted in Vero E6 cells in triplicates. The supernatants were diluted 10 folds from 10^−1^ to 10^−5^ with DMEM without serum. Confluent monolayer Vero E6 cells were prepared a day prior to the plaque assay by seeding 2.5 × 10^5^ of Vero E6 cells in 12-well plates and culturing them with DMEM supplemented with 10% FBS. On the day of the plaque assay, the growth medium was removed, and the cells were infected with 0.3 mL of 10^−1^- to 10^−5^-diluted samples. A mock infection was performed using only DMEM without serum and acted as the negative control. The cells were incubated at 37 °C, 5% CO_2_ for 60 min. After incubation, the virus inoculum was removed, and the cell monolayer was washed and then covered with 1 mL of overlay medium (0.8% *w*/*v* agarose: DMEM; 1:3) and allowed to solidify at room temperature. The plates were incubated at 37 °C in 5% CO_2_ for 3 to 5 days before fixation using 4% paraformaldehyde phosphate buffer solution (Nacalai Tesque, Kyoto, Japan) for 2 h. The overlay agar was removed under running water and the cell monolayer was stained with 0.5% crystal violet in 25% methanol. Clear plaques were quantified to determine the plaque-forming units per unit volume (PFU/mL). The plaque assay was performed in triplicates, both technically and biologically.

### 4.4. Infectivity Assay

Vero E6 cells were seeded in 6-well plates at 5.0 × 10^5^ cells/well in DMEM supplemented with 10% FBS a day prior to the infection. The cells were infected with SARS-CoV-2 Alpha, Beta, Delta, and Omicron variants at an MOI of 0.001 for 60 min, with occasional shaking every 10 to 15 min. After adsorption, the media was removed and the cells were washed with phosphate-buffered saline (PBS; Gibco, Paisley, UK). Then, 2 mL of fresh DMEM supplemented with 2% FBS was added to each well. The presence of cytopathic effects (CPEs) was observed daily for 5 days. The supernatant was collected at five different time points, i.e., 0, 24, 48, 72, and 96 h.p.i for investigating virus growth kinetics. Approximately 200 mL of the supernatants was aliquoted at the designated time points and heat-inactivated for 1 h. The remaining supernatants served as the infectious culture supernatants. All samples were stored at −80 °C until further analyzed in the virus titration assay and viral RNA extraction. All infections were done in triplicates, technically and biologically.

### 4.5. Ribonucleic Acid (RNA) Extraction and Nucleic Acids Amplification

The total RNA in the culture supernatant was extracted using the QIAamp viral RNA Mini Kit (Qiagen, Hilden, Germany) and the presence of the virus genome was evaluated using qRT-PCR by employing the BGI’s Real-Time Fluorescent RT-PCR kit (BGI, Wuhan, China). The amplification was based on the SARS-CoV-2 ORF1ab and N genes.

### 4.6. Whole Genome Sequencing

The nucleic acid sequences of the variants were sequenced to determine their genomic stability after three passages. The viral RNA was extracted and amplified as described before. Next-generation sequencing (NGS) was performed by using the Oxford Nanopore GridION system (Apical Scientific, Seri Kembangan, Malaysia). Four data sets (Alpha variant, Beta variant, Delta variant, and Omicron variant) were obtained and analyzed using the Nextclade Web v3.8.2 (https://clades.nextstrain.org, accessed on 10 July 2024)).

### 4.7. Statistical Analysis

Data values represent the mean ± SD from 3 independent experiments. The statistical analysis was performed using two-way ANOVA and Tukey’s multiple comparison test to compare the levels of gene expression and viral titers of four variants at the designated time points. A *p*-value < 0.05 was considered significant for all statistical tests. All of the data were analyzed using the GraphPad Prism 9.0 (GraphPad Software, San Diego, CA, USA).

## 5. Conclusions

Overall, the replication kinetics of Alpha, Beta, Delta, and Omicron variants in Vero E6 cells exhibited similar growth patterns. Among the variants, the Beta variant showed the most efficient replication; nonetheless, the Omicron variant displayed a relatively slower replicative ability than its fraternity variants, which is most likely due to its low fusogenic properties. In conclusion, compared to the Omicron variant, the earlier VOCs showed minor variations in their replication kinetics in Vero E6 cells. This, therefore, implies the importance of monitoring the replicative abilities of the circulating and upcoming variants, in addition to their transmissibility, pathogenicity, and immune escape capabilities, in promoting public health measures to control viral spread. It is also pivotal to monitor genetic variations in the variants, especially in the S1/S2 region of the spike before and after propagating the SARS-CoV-2 variants in vitro to ensure the genome stability of the working stocks. In the case of viral genome mutations, their impacts on virus replication, infectivity, and shedding can be further explored using omics approaches.

## Figures and Tables

**Figure 1 ijms-25-10541-f001:**
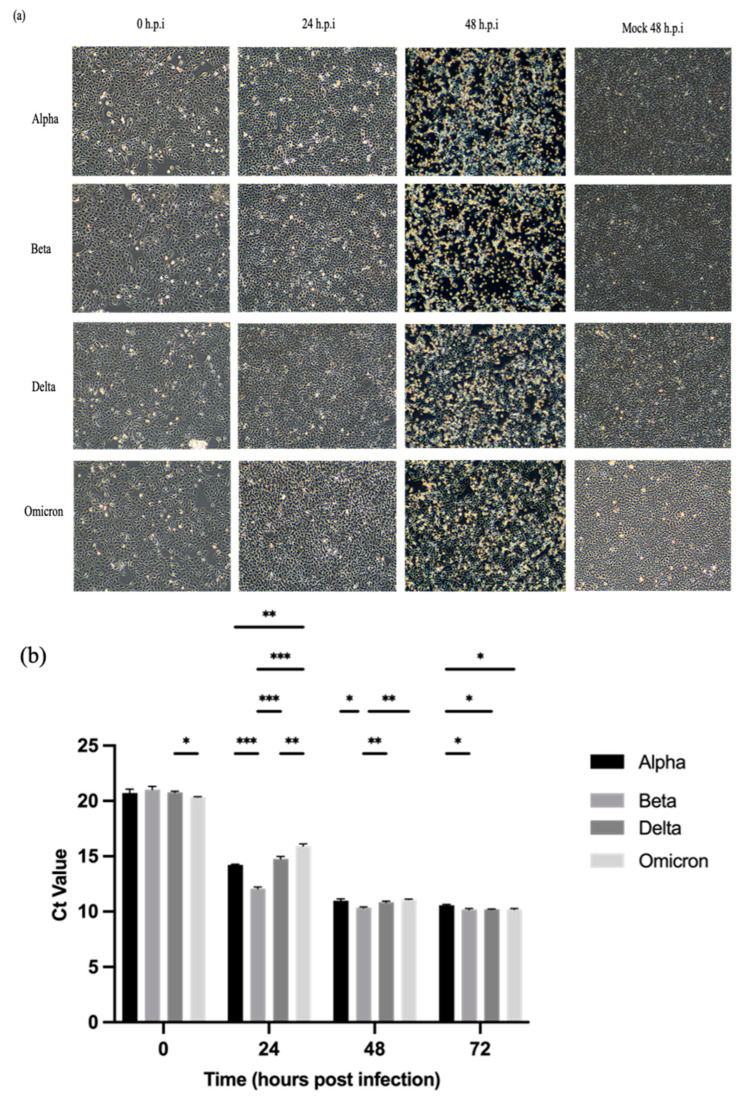
Cytopathic effects (CPEs) in Vero E6 cells inoculated with SARS-CoV-2 variants at MOI of 0.001 and detection of SARS-CoV-2 ORF1ab gene in the culture supernatants at the designated time points using qRT-PCR. (**a**) Representative images of infected Vero E6 cells. The CPEs were remarkably evident at 48 h.p.i. for all variants. The morphology changes of infected cells were visualized at 100× magnification. (**b**) Comparison of Ct values at the designated time points. All four variants exhibited almost similar trends in RNA multiplication. At 24 h.p.i, the Beta variant showed the lowest Ct value followed by the Alpha and Delta variants, the Omicron variant demonstrated the highest Ct value among the variants. The Ct values plateaued after 48 h.p.i. The data values represent the mean ± SD derived from three independent experiments. The statistical analysis was performed with the two-way ANOVA and Tukey’s multiple comparison tests. * *p* < 0.033, ** *p* < 0.002, and *** *p* < 0.001.

**Figure 2 ijms-25-10541-f002:**
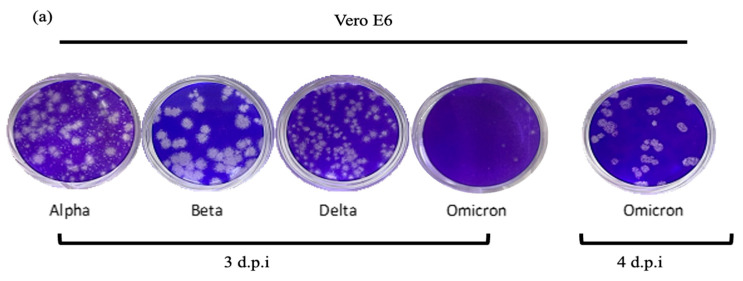
The morphology of plaques and titration of the Alpha, Beta, Delta, and Omicron variants in the Vero E6 cell line. (**a**) Representative images of the plaque assay. The Beta variant formed larger clear zones than the Alpha, Delta, and Omicron variants. The formation of visible plaques by the Omicron variant was delayed and the plaques could only be enumerated on day-4 post-infection (d.p.i). (**b**) Comparison of the virus titers of the SARS-CoV-2 variants (PFUs/mL). The supernatant was harvested at the designated time points and the virus titers of the samples were determined using the plaque assay. The titers of all four variants peaked at 48 h.p.i with the Beta variant showing the highest titer. At 24 h.p.i, the Beta variant produced a significantly higher level of infectious virions than the others. The data values represent the mean ± SD derived from three independent experiments. The statistical analysis was performed with the two-way ANOVA and Tukey’s multiple comparison tests. * *p* < 0.033, ** *p* < 0.002, and *** *p* < 0.001.

**Figure 3 ijms-25-10541-f003:**
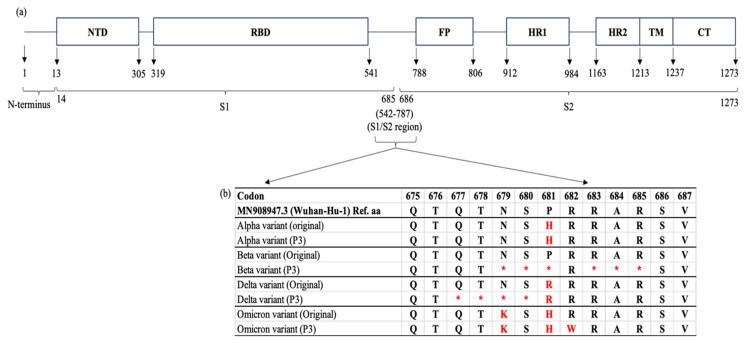
A schematic view of the SARS-CoV-2 spike (S) gene and the locations of mutations after cell culture adaptations. (**a**) The S protein consists of 1273 amino acid (aa): 1–13 aa (a single peptide located at the N-terminus), 14–685 aa (S1 subunit), and 686–1273 aa (S2 subunit). Generally, the S1 and S2 domains are responsible for the receptor binding and membrane fusion of SARS-CoV-2 virus into the host cell, respectively. The whole-genome sequencing revealed that the Beta, Delta and Omicron variants possessed mutations in the S1/S2 region when compared to their original sequences [Alpha (B.1.1.7, EPI-ISL-877228), Beta (B.1.351, EPI-ISL-4730383), Delta (AY.59, EPI-ISL-3425462), and Omicron (BA.1.1, EPI-ISL-11105858)]. (**b**) The Beta and Delta variants were found to carry deletion mutations at the multibasic cleavage site [^677^QTNSPRRAR^685^]. The amino acid substitution was reported in the Omicron (R682W) variant. NTD: N terminal domain, RBD: receptor binding domain, FP: fusion peptide, HR1: heptapeptide repeat sequence 1, HR2: heptapeptide repeat sequence 2, TM: TM domain, CT: cytoplasm domain. * aa deletion. The red asterisks and alphabet indicate aa deletions and substitution, respectively in comparision to the original strains.

**Table 1 ijms-25-10541-t001:** A list of unique mutations, either by amino acid (aa) substitution or deletion, found in the S protein of each variant after three cycles of passaging (P3) in comparison to their original sequences. The Beta and Delta variants showed aa deletion mutations (in bold), whereas the Omicron variant demonstrated an aa substitution (in bold). The Alpha variant maintained its genome stability throughout the passaging in the Vero E6 cell line.

Variant	Mutations	
	aa Substitution/ADDITION	aa Deletion
Alpha (original)	N501Y, A570D, D614G, P681H, T716I, S982A, D1118H	H69-, V70-, Y144-
Alpha (P3)	N501Y, A570D, D614G, P681H, T716I, S982A, D1118H	H69-, V70-, Y144-
Beta (original)	A27S, D80A, D215G, K417N, E484K, N501Y, D614G, A701V	L241-, L242-, A243-
Beta (P3)	A27S, D80A, D215G, K417N, E484K, N501Y, D614G, A701V	L241-, L242-, A243-, **N679-**, **S680-**, **P681-**, **R683-**, **A684-**, **R685-**
Delta (original)	T19R, G142D, R158G, A222V, L452R, T478K, D614G, P681R, D950N	E156-, F157-
Delta (P3)	T19R, G142D, R158G, A222V, L452R, T478K, D614G, P681R, D950N	E156-, F157-, **Q677-**, **T678-**, **N679-**, **S680-**
Omicron (original)	A67V, T95I, Y145D, L212I, G339D, R346K, S371L, S373P, S375F, K417N, N440K, G446S, S477N, T478K, E484A, Q493R, G496S, Q498R, N501Y, Y505H, T547K, D614G, H655Y, N679K, P681H, N764K, D796Y, N856K, Q954H, N969K, L981F	H69-, V70-, G142-, V143-, Y144-, N211-
Omicron (P3)	A67V, T95I, Y145D, L212I, G339D, R346K, S371L, S373P, S375F, K417N, N440K, G446S, S477N, T478K, E484A, Q493R, G496S, Q498R, N501Y, Y505H, T547K, D614G, H655Y, N679K, P681H, **R682W**, N764K, D796Y, N856K, Q954H, N969K, L981F	H69-, V70-, G142-, V143-, Y144-, N211-

## Data Availability

Dataset available on request from the authors.

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
