# Peer review of "Insights into the Replication Kinetics Profiles of Malaysian SARS-CoV-2 Variant Alpha, Beta, Delta, and Omicron in Vero E6 Cell Line"

_ijms, 2024, doi:10.3390/ijms251910541_

Round 1

Reviewer 1 Report

Comments and Suggestions for Authors

Please state the number of independent experiments for the plaque assay.

Why was SARS-CoV-2 gamma not investigated? Please add this information.

Please add a picture of SARS-CoV-2 Omicron plaque on day 3.

Please, discuss your results in the light of the avirulence theory and virulence transmission-trade-off theory.

A deeper discussion on how the cell culture adaptations might influenza viral load/plaque size and therefore lead to an overestimation of transmissibility.

Comments on the Quality of English Language

-

Author Response

1. Summary

2. Questions for General Evaluation

Reviewer’s Evaluation

Response and Revisions

Does the introduction provide sufficient background and include all relevant references?

Can be improved

Refer to the point-by-point response.

Are all the cited references relevant to the research?

Can be improved

Refer to the point-by-point response.

Is the research design appropriate?

Can be improved

Refer to the point-by-point response.

Are the methods adequately described?

Can be improved

Refer to the point-by-point response.

Are the results clearly presented?

Yes

Are the conclusions supported by the results?

Yes

3. Point-by-point response to Comments and Suggestions for Authors

Comments 1:

Please state the number of independent experiments for the plaque assay.

Response 1:

The manuscript has been revised, the repeat number of the plaque assay is stated in the Methods section (Lines 417-418). “The plaque assay was repeated in triplicates; technically and biologically”.

Comments 2:

Why was SARS-CoV-2 Gamma not investigated? Please add this information.

Response 2:

The gamma variant was not included in the study because it was not one of the circulating variants in Malaysia. The information is added into the Discussion (line 212-213).

Comments 3:

Please add picture of SARS-CoV-2 Omicron plaque on day 3.

Response 3:

The figure of plaques formed by the Omicron variant is now added in Figure 2 (a). The plaque morphology was mostly invisible on day-3 p.i. Clear and discreet plaques were only observed on day-4 p.i. The newly added information can be found in Figure 2 (a), and in lines 146-147.

Comment 4:

Please discuss your results in the light of the avirulence theory and virulence transmission-trade-off theory.

Response 4:

The points are now included in the revised manuscript (lines 239-247).

Comment 5:

A deeper discussion on how cell culture adaptations might influence viral load/plaque size and therefore lead to an overestimation of transmissibility.

Response 5:

The comment is now addressed and a short paragraph relating cell culture adaptation to an overestimation of viral transmissibility is included in the text (lines 366-374).

4. Response to Comments on the Quality of English Language

None

Reviewer 2 Report

Comments and Suggestions for Authors

The manuscript entitled “Insights into the Replication Kinetics Profiles of Malaysian SARS-CoV-2 Variant Alpha, Beta, Delta, and Omicron in Vero E6 cell line” has been submitted to IJMS by Zawawi and colleagues.  This study analyse and compare the replication of different SARS-CoV-2 variants in VERO E6 cell.  Overall, while these results are not entirely novelty, they mostly appear rigorous and will advance the field.  Some comments are noted below:

-In figure 1 and 2, the sections b and c, seem redundant as they portrait the same data simply using a different graph format. I would advise the use of one or the other.

-In figure 1 b/c, data is reported up to 72 hours post infection, while in figure 2 b/c is up to 96. Why is there no data at 96 hours for figure 1? I think it could polish the data presented and making it more readable and comparable for the readers.

-the authors use the Plaque formation assay to define the viral titre, but in the discussion compare their results to Mautner et al. (2022), who used the TCID50/mL for titre determination. The authors should determine also their TCID50/mL, this not only would allow for a direct comparison to other works already published, but would also give information on the relation between PFU/mL and TCID50/mL as those values sometimes appear to not be directly comparable.

Lastly, I have a small consideration. The genomic analysis of the SARS-CoV-2 strains before and after propagation, adds novelty to the work. It would be interesting to have data on the replication kinetics of the original strains without the aa/del changes. I do understand that probably the viral titre wasn’t enough high to perform the experiments, but I will suggest for future works the possibility of expanding on the role of the mutation identified.

Author Response

1. Summary

2. Questions for General Evaluation

Reviewer’s Evaluation

Response and Revisions

Does the introduction provide sufficient background and include all relevant references?

Yes

Refer to the point-by-point response.

Are all the cited references relevant to the research?

Yes

Is the research design appropriate?

Yes

Are the methods adequately described?

Yes

Are the results clearly presented?

Yes

3. Point-by-point response to Comments and Suggestions for Authors

Comments 1:

In Figure 1 and 2, the sections b and c, seem redundant as they portrait the same data simply using a different graph format. I would advise the use of one or the other.

Response 1:

Figures 1 and 2 have been amended, and only the bar charts with statistical comparison data between variants are remained to represent the intended analyses.

Figure 1 (b), page 4, lines 111-112.

Figure 2 (b), page 5, lines 148-149.

Comments 2:

In figure 1 b/c, data is reported up to 72 hours post infection, while in figure 2 b/c is up to 96. Why is there no data at 96 hours for figure 1? I think it could polish the data presented and making it more readable and comparable for the readers.

Response 2:

According to Fig 1 b (then 1c), the viral RNA genome multiplication became plateau after 48 and 72 h.p.i. Therefore, monitoring the multiplication thresholds beyond 72 hpi would not provide additional information. In Figure 2, the changes in the virus titers were described up to 96 h.p.i to better decipher the replication patterns of the variants. Further explanation is provided in the Discussion, pg. 9, lines 217-223.

Comments 3:

The authors use the plaque formation assay to define the viral titers, but in the discussion compare their results to Mautner et al. (2022), who used the TCID50/mL for titers determination. The authors should determine also their TCID50/mL, this not only would allow for a direct comparison to other works already published but would also give information on the relation between PFus/mL and TCID50/mL as those values sometimes appear to not be directly comparable.

Response 3:

To avoid confusion in readers, the reference is removed and the discussion is modified to better fit the context.

Comment 4:

Lastly, I have small consideration. The genomic analysis of the SARS-CoV-2 strains before and after propagation adds novelty to the works. It would be interesting to have data on the replication kinetics of the original strains without the aa/del changes. I do understand that probably the viral titre wasn’t enough high to perform the experiments, but I will suggest for future works the possibility of expanding on the role of mutation identified.

Response 4:

The suggestion of employing omics approaches to determine the effects of viral genome mutations is included in the Conclusion (lines 463-464).

4. Response to Comments on the Quality of English Language

None

Round 2

Reviewer 1 Report

Comments and Suggestions for Authors

Accept as is. no more comments.